# An Iterative Deconvolution-Time Reversal Method with Noise Reduction, a High Resolution and Sidelobe Suppression for Active Sonar in Shallow Water Environments

**DOI:** 10.3390/s20102844

**Published:** 2020-05-16

**Authors:** Chun-Xiao Li, Ming-Fei Guo, Hang-Fang Zhao

**Affiliations:** 1College of Mechanical Engineering, Zhejiang University of Technology, Hangzhou 310023, China; gmf@zjut.edu.cn; 2Key Laboratory of Special Purpose Equipment and Advanced Processing Technology, Ministry of Education and Zhejiang Province, Zhejiang University of Technology, Hangzhou 310023, China; 3College of Information Science and Electronic Engineering, Zhejiang University, Hangzhou 310027, China; hfzhao@zju.edu.cn; 4Key laboratory Ocean Observation-Imaging Testbed of Zhejiang Province, Zhoushan 316021, China

**Keywords:** matched filter, time reversal, iterative deconvolution, target detection, active sonar, shallow water environments, cross-ambiguity function

## Abstract

Matched filtering is widely used in active sonar because of its simplicity and ease of implementation. However, the resolution performance generally depends on the transmitted waveform. Moreover, its detection performance is limited by the high-level sidelobes and seriously degraded in a shallow water environment due to time spread induced by multipath propagation. This paper proposed a method named iterative deconvolution-time reversal (ID-TR), on which the energy of the cross-ambiguity function is modeled, as a convolution of the energy of the auto-ambiguity function of the transmitted signal with the generalized target reflectivity density. Similarly, the generalized target reflectivity density is a convolution of the spread function of channel with the reflectivity density of target as well. The ambiguity caused by the transmitted signal and the spread function of channel are removed by Richardson-Lucy iterative deconvolution and the time reversal processing, respectively. Moreover, this is a special case of the Richardson-Lucy algorithm that the blur function is one-dimensional and time-invariant. Therefore, the iteration deconvolution is actually implemented by the iterative temporal time reversal processing. Due to the iterative time reversal method can focus more and more energy on the strongest target with the iterative number increasing and then the peak-signal power increases, the simulated result shows that the noise reduction can achieve 250 dB in the “ideal” free field environment and 100 dB in a strong multipaths waveguide environment if a 1-ms linear frequency modulation with a 4-kHz frequency bandwidth is transmitted and the number of iteration is 10. Moreover, the range resolution is approximately a delta function. The results of the experiment in a tank show that the noise level is suppressed by more than 70 dB and the reverberation level is suppressed by 3 dB in the case of a single target and the iteration number being 8.

## 1. Introduction

Matched filtering (MF) is widely used in active sonar because of its simplicity and ease of implementation [1,2,3,4]. The purpose of active sonar is not only to detect the echo buried in the noise/reverberation but also to estimate the range of the target and distinguish the target echo from clutters to avoid false alarms. In order to achieve the purpose, the sonar system must not only provide a sufficient processing gain to combat the noise but also exhibit a good time and frequency resolution. Moreover, the sidelobe level is another important factor. If the sidelobe is relatively high, the sidelobes of a larger response can bury the mainlobe of a smaller response located nearby. Therefore, the capability of noise reduction, range sidelobe level and range resolution are the three key issues considered by active radar/sonar. The usual way to solve the problems of high sidelobe level or low resolution is radar/sonar waveform design [5,6,7,8,9,10,11]. Generally speaking, different waveforms have different resolution performances. For example, the pulsed continuous wave has a high Doppler resolution on the order of 1/T (T is duration) but lacks range resolution, whereas the linear frequency modulation (LFM) signal has a high range resolution, depending on the bandwidth, but is insensitive to Doppler. Therefore, multiple-input-multiple-output (MIMO) radar systems have been intensively studied in the recent decade, which simultaneously transmit multiple waveforms. After years of research, great progress has been made in MIMO radar waveform design. For the current state of art, sidelobe suppression merely by means of waveform design seems to achieve a limit [11]. Therefore, other approaches, such as the coherent CLEAN algorithm [12] and coherence factor filtering [13,14] have been reported to suppress the sidelobe.

Since it is not meaningful to design waveforms to suppress additive thermal noise, the high gain is usually provided by a high product of the signal bandwidth and duration. For a peak-power-limited sonar system, the improvement in detection performance is mainly reliant on increasing the signal energy/duration under noise-limited detection conditions. The characteristic of rapid time variation in underwater environments may not be suited to the use of signals with a long duration. The other way to combat noise is the design of image reconstruction, which perform Wiener filtering [15] or rational orthogonal wavelet analysis filter bank [16] to suppress noise in active radar on the basis of MIMO waveform design. The shallow water environment is a complicated media with the sea surface upward and the seafloor downward [17,18]. For active sonar in shallow water environments, since the received signal is distorted due to the time-delay and Doppler spread caused by the channel, the received signal comprises the ambiguity of both the transmitted signal and the channel. Therefore, the difficulty of detecting a target is greatly exemplified when a low Doppler target is situated in shallow water environments. It cannot exploit the difference in Doppler between a moving target and the bottom clutter to separate the target echo from the clutter. Analysis based on experimental data demonstrated that a moving target can be “unambiguously” identified in the high Doppler space due to the absence of a high-Doppler background [19,20].

The objective of this paper is to propose an alternative way to provide the active sonar with a superior processing gain, super-resolution and low-level sidelobe. Recent publications have shown that deconvolution-based methods can yield higher resolution and higher processing gain than the conventional methods, such as the CLEAN algorithm [12], Wiener filter [16] and Richardson-Lucy algorithm [21,22,23,24,25]. The Richardson-Lucy algorithm was devised by Richardson [26] and Lucy [27], which is an iterative deconvolution algorithm and recently applied in active radar/sonar to remove the blur caused by the transmitted signal [23,24,25]. In [23,24], the Richardson-Lucy algorithm is applied to the range-Doppler image, whereas it is used to the range image in the case of zero-Doppler in the paper. It will be shown that the iterative deconvolution is actually an iterative time reversal processing in the special case of a one-dimensional, time-invariant blur function, which is easy to be met for the case in the paper. The traditional iterative time reversal is achieved using an array name time reversal mirror [28,29,30,31,32,33,34,35]. This is the first time that the time reversal is implemented in the time domain. Compared with the usual Richardson-Lucy, the special case has many advantageous characteristics. The speed of iterative deconvolution in the paper is much faster than the usual one whose iterative speed is lower than the Wiener filter [23]. Moreover, due to the iterative time reversal can focus more and more energy on the strongest target, the peak-signal power of the iterative deconvolution is higher and higher with the number of iteration increasing. Therefore, the processing gain to combat noise and the peak-to-sidelobe level are greatly improved. Instead of transmitting multiple waveforms [5,6,7,8,9,10,11,12,13,14,15,16], the transmitted signal is a single pulse of LFM in the paper.

The iterative deconvolution can only remove the ambiguity caused by the transmitted signal. Moreover, the knowledge of the spread function of channel cannot be precisely known in practice and then is estimated in [24,25]. In this paper, we put forward an iterative deconvolution-time reversal (ID-TR) method that can also remove the blur caused by the channel using the time-reversal method [28,29,30,31,32,33,34,35]. It is shown below with simulated and real data that the proposed method can not only provide a high resolution beyond Rayleigh resolution limit, a super processing gain that can combat noise and a low sidelobe but also discriminate the target from reverberation to a certain degree under reverberation-limited condition.

This paper is organized as follows. In Section 2, we review the received signals in free environments and in waveguide environments, and model the cross-ambiguity function as a convolution of auto-ambiguity function and the generalized reflectivity density function of the target, and finally put forward the ID-TR method. Section 3 presents the simulation results of target detection using ID-TR and MF in free field and waveguide environments. Section 4 shows the experimental results in a tank. The discussion and conclusion are given in Section 5.

## 2. Iterative Deconvolution-Time Reversal (ID-TR) Method

### 2.1. Modeling of the Received Signal

In a free field environment, the signal *r*(*t*) reflected by a single target can be modeled as:(1)r(t)=ρs(αt−τ)+n(t),
where *s*(*t*) and *n*(*t*) are transmitted signal and noise, respectively, *ρ* is amplitude attenuation and phase variation, depending on the target reflectivity coefficient and acoustic propagation characteristic, and *α* and *τ* are the Doppler and delay, respectively. The proposed method developed for the situation that ignores the Doppler, can be easy to extend to the moving target as in [24].

The received signals of multiple incoherent targets in underwater multipath channel, can be expressed as [4]:(2)$$r(t)=\sum_{m=1}^{M}\sum_{k=1}^{K}\rho_{mk}s(t-\tau_{mk})+n(t)=\rho (t)⊗s(t)+n(t),$$
where *M* and *K* are the number of targets and that of the paths, respectively, $\rho_{mk}$, $\tau_{mk}$ are the variation of amplitude and phase, delay for the *m*-th target and *k*-th path, respectively, which are dependent on the parameters of the target and the channel, and ⊗ is linear convolution.

Since the characteristic of underwater channel is that it bounded in a vertical direction and unbounded in a horizontal direction, the sound propagation should take into account the multipath caused by the reflection from the upper and lower boundary and the refraction by the inhomogeneity of channel. Consider the procedure of active sonar. Provided that the active sonar has a single source element and a single receiver element, *ρ*(*t*) should include the forward propagation from the source to target, the target reflectivity function, and the backward propagation from the target to the receiver:(3)$$\rho (t)=h_{f}(t)⊗c(t)⊗h_{b}(t)=h(t)⊗c(t),$$
where *h_f_*(*t*), *h_b_*(*t*) are the spread function of the forward and backward propagation channels, respectively, *h_f_*(*t*) = *h_b_*(*t*), $h(t)=h_{f}(t)⊗h_{b}(t)$, for the static waveguide considered in the paper. The spread function of channel is modeled as the weighted sum of multipath signal. The target reflectivity function consists of the reflectivity parameters of multiple targets.
(4)$$\begin{array}{l} h(t)=\sum_{k=1}^{K}h_{k}\delta (t-\tau_{k})\\ c(t)=\sum_{m=1}^{M}c_{m}\delta (t-\tau_{m}) \end{array}$$

Therefore, *ρ*(*t*) in Equation (2) should comprise of the characteristics of both target reflection and channel propagation.
(5)$$\rho (t)=h(t)⊗c(t)=\sum_{m=1}^{M}\sum_{k=1}^{K}h_{k}c_{m}\delta (t-\tau_{k}-\tau_{m})$$
*ρ*(*t*) includes the effect of the spread function of channel *h*(*t*) and the reflectivity density function of the target *c*(*t*) and is named as generalized reflectivity density function of the target.

Setting *ρ_mk_ = h_k_c_m_*, *τ_mk_* = *τ_m_* + *τ_k_*, we can see that *ρ*(*t*) in Equation (5) is equals to that of Equation (2). The free field environment can be treated as a special waveguide environment. The spread function of channel has only direct path, that is *K* = 1, and becomes *h*(*t*) = *h*δ(*t* − *τ*). Therefore, the corresponding generalized target reflectivity density function for a single target is:(6)ρ(t)=hcδ(t−τ)=ρδ(t−τ)

One finds that Equation (1) can be obtain if we substitute Equation (6) into Equation (2) with *α* = 1.

### 2.2. Matched Filter

A sonar/radar uses the delay or the Doppler of the received waveforms as a means of obtaining surveillance information, and requires the use of waveforms that are carefully designed to provide adequate resolution and avoid ambiguity. The major analytical tool used to design such waveforms is the ambiguity function.

The auto-ambiguity function of the transmitted signal *s*(*t*) with finite energy is defined as [19]:(7)$$\chi (\tau ,\alpha )=\int_{-\infty }^{\infty }s(t)s_{\alpha }^{*}(t-\tau )dt$$
where superscript * is conjugation, and $s_{\alpha }(t)=\sqrt{\alpha }s(\alpha t)$ is the time compressed or dilated version of the transmitted signal *s*(*t*), where α is the compression/dilation ration, referred as the Doppler ratio. One sees that the Doppler and range resolution is determined predominantly by the ambiguity function of the transmitted signal. The wideband cross-ambiguity function of the received signal is a superposition of the wideband auto-ambiguity functions of the transmitted signal delayed by the multipath arrival time and weighted by the multipath amplitude:(8)$$\chi_{c}(\tau ,\alpha )=\int_{-\infty }^{\infty }r(t)s_{\alpha }^{*}(t-\tau )dt,$$

For a given Doppler ratio, Equation (8) can be viewed as a matched filter processor, or a correlation receiver, where *s_α_*(*t*), referred to as the replica signal, is used to match (correlate) with the data. We can estimate the range and the Doppler using the matched filter. However, delay and Doppler resolutions are dependent on the main lobe of auto-ambiguity function. Moreover, the gain of the matched filter is also limited by the transmitted signal.

### 2.3. Convolution Model of Wideband Cross-Ambiguity Function

In order to guarantee their non-negativities, in this paper, we chose the energy of auto-ambiguity function and the cross-ambiguity and set *α* = 1 by ignoring the Doppler. Equations (7) and (8) can be expressed as:(9)$${\left|\chi (\tau )\right|}^{2}={\left|\int_{-\infty }^{\infty }s(t)s^{*}(t-\tau )dt\right|}^{2},$$
(10)$${\left|\chi_{c}(\tau )\right|}^{2}={\left|\int_{-\infty }^{\infty }r(t)s^{*}(t-\tau )dt\right|}^{2}$$

Substituting the received signal Equation (2) into Equation (10), we can get the expectation of it as: (11)$$\begin{array}{l} {\left|\chi_{c}(\tau )\right|}^{2}={\left|\int_{-\infty }^{\infty }\sum_{m=1}^{M}\sum_{k=1}^{K}\rho_{mk}s(t-\tau_{mk})s^{*}(t-\tau )dt\right|}^{2}={\left|\sum_{m=1}^{M}\sum_{k=1}^{K}\rho_{mk}\int_{-\infty }^{\infty }s(t-\tau_{mk})s^{*}(t-\tau )dt\right|}^{2}\\ ={\left|\sum_{m=1}^{M}\sum_{k=1}^{K}\rho_{mk}\chi (\tau -\tau_{mk})\right|}^{2}\neq {\left|\rho (\tau )\right|}^{2}⊗{\left|\chi (\tau )\right|}^{2} \end{array}$$

If and only if it is in the free field environment ${\left|\sum_{m=1}^{M}\sum_{k=1}^{K}\rho_{mk}\chi (\tau -\tau_{mk})\right|}^{2}={\left|\rho (\tau )\right|}^{2}⊗{\left|\chi (\tau )\right|}^{2}$. For simplification, we ignore the absolute value sign ${\|}^{2}$ in the following derivation process. 

### 2.4. Iterative Deconvolution-Time Reversal Method

The question is how to estimate the generalized target reflectivity density function in Equation (11). This is a deconvolution process with the prior knowledge including the sampling cross-ambiguity function *χ_c_*(*τ*) and the auto-ambigity function *χ*(*τ*) of the transmitted signal. 

The estimation of an image from noisy data can be treated by the principle of maximum likelihood. The Richardson–Lucy deconvolution algorithm is derived by Richardson and Lucy from Bayes formula of probability theory. In this paper, the Richardson-Lucy algorithm is developed using information theory [36], which minimize a discrepancy measurement named Csiszár discrimination
(12)$$L\left(a(x),b(x)\right)=\int_{-\infty }^{\infty }a(x)\log\frac{a(x)}{b(x)}dx-\int_{-\infty }^{\infty }[a(x)-b(x)]dx,$$
where *a*(*x*) and *b*(*x*) are elements of the space of nonnegative real functions. The Csiszár discrimination is an appropriate way to measure the separation between nonnegative functions.

The energies of *χ_c_*(*τ*) and *χ*(*τ*) are nonnegative. At the same time, we can ensure that the integral of *χ_c_*(*τ*) and *χ*(*τ*) are one by rescaling *ρ*(τ). The estimation of cross-ambiguity function is: (13)$$E[{\hat{\chi }}_{c}(\tau )]=\int_{-\infty }^{\infty }\rho (v)\chi (\tau |v)dv.$$
where χ(τ|v) becomes the impulse response function of the filter that need not be time invariant in general.

The task is that we find a nonnegative function *ρ*(*v*) that integrates to one and minimizes the discrimination [37]:(14)$$L\left(\chi_{c}(\tau ),E[{\hat{\chi }}_{c}(\tau )]\right)=L\left(\chi_{c}(\tau ),\int_{-\infty }^{\infty }\rho (v)\chi (\tau |v)dv\right)=\int_{-\infty }^{\infty }\chi_{c}(\tau )\log\frac{\chi_{c}(\tau )}{\int_{-\infty }^{\infty }\rho (v)\chi (\tau |v)dv}d\tau$$

The Csiszár discrimination is nonnegative and equal to zeros if and only if $\chi_{c}(\tau )=E[{\hat{\chi }}_{c}(\tau )]$ [38]. The function that achieves the minimum is written as:(15)$$\hat{\rho }\left(v\right)=\arg\min_{\rho (v)}\int_{-\infty }^{\infty }\chi_{c}(\tau )\log\frac{\chi_{c}(\tau )}{\int_{-\infty }^{\infty }\rho (v)\chi (\tau |v)dv}d\tau$$

Any solution that achieves the desired minimum must satisfy the Kuhn-Tucker condition [38]:(16)$$\int_{-\infty }^{\infty }\frac{\chi (\tau |v)}{\int_{-\infty }^{\infty }\rho (v)\chi (\tau |v)dv}\chi_{c}(\tau )d\tau =\{\begin{array}{c} =H_{0}(v)\\ \leq H_{0}(v) \end{array},$$
with equality at all *v* for which *ρ*(*v*) is nonnegative and nonzero, where $H_{0}(v)=\int_{-\infty }^{\infty }\chi (\tau |v)dv$, which equals to one by rescaling *ρ*(τ). 

It is difficult to find out its analytical solution, whereas the iterative solution can be obtained by multiplying through by *ρ*(*v*) in Equation (16) with equality at all *v*:(17)$$\rho^{\left(r+1\right)}\left(v\right)\;=\rho^{\left(r\right)}\left(v\right)\int_{-\infty }^{\infty }\frac{\chi (\tau |v)}{\int_{-\infty }^{\infty }\chi (\tau |v)\rho^{\left(r\right)}(v)dv}\chi_{c}(\tau )d\tau ,$$
where *r* is the number of iteration. The recursion is the expression of the Richardson-Lucy algorithm. It was proved that the Kullback discrimination (that is Csiszár discrimination in information theory) of the Richardson-Lucy algorithm is monotonically decreasing with the number of iterations increasing. Finally, the solution must converge to the desired function [38]. 

In the special case of a one-dimensional, time-invariant blur function, the integrals become convolutions and Equation (17) can be expressed as:(18)$$\rho^{\left(r+1\right)}\left(\tau \right)\;=\rho^{\left(r\right)}\left(\tau \right)\frac{\chi (\tau )}{\chi (\tau )⊗\rho^{\left(r\right)}\left(\tau \right)}⊗\chi_{c}(-\tau )$$

In this paper, this situation can be met as shown in Equation (11) and then the iterative deconvolution is achieved using Equation (18). This is a similar procedure as iterative time reversal processing. However, the traditional iterative time reversal is achieved using an array named time reversal mirror. Now the time reversal is implemented not in space but in the time domain. Compared to the general Richardson-Lucy algorithm, the special case has a superior ability of anti-noise and a rapid speed of convergence. 

However, the knowledge of multipath channel cannot be precisely known in practice. The channel function is estimated in [24,25]. Provided that the number of iterations is *R*, the final solution of generalized target reflectivity density function is $\rho^{R}(v)$, which can be expressed as the convolution of the spread function of channel *h*(*v*) and the reflectivity density function of the target *c*(*v*) such as in Equation (5):(19)$$\rho^{R}(v)=h(v)⊗c(v)$$

To emphasize the impact of the channel propagation effects (multipath) induced by the clutter and to keep the focus on the role of time reversal on detection, the extreme case of either a single antenna in the monostatic context or a single transmitting antenna and a single receiving antenna in the bistatic problem is considered [35]. In this paper, to remove the blur caused by the channel propagation, we use the time reversal method with a single antenna based on the concept that time reversal is equivalent to phase-conjugation in the frequency domain where the power detector analysis will be carried out [35,39]. The generalized target reflectivity density function is transformed into Fourier in a short time window between (*v*, *v* + Δ*τ*):(20)$$\rho^{R*}(f,v)=c^{*}(f,v)h^{*}(f,v).$$

The traditional time reversal method is that the signal is retransmitted into the real or modeled channel to focus on the target and then obtain the reflectivity density function of the target, which is called time reversal or numerical time reversal method [28,29,30,31,32,33,34]. As mentioned above, it is difficult to know the channel function. We achieve the time reversal process by multiplying $\rho^{R}(f,v)$ with its own conjugation, which is a power detector:(21)$${\left|\rho^{R}(f,v)\right|}^{2}=\rho^{R*}(f,v)\rho^{R}(f,v)=c^{*}(f,v)h^{*}(f,v)h(f,v)c(f,v)={\left|c(f,v)\right|}^{2},$$
where ${\left|h(f,v)\right|}^{2}=1$ provided that the channel function is normalized. In this case, the reflectivity energy density function of the target can be obtained. In fact, because the final result obtained by iterative deconvolution is not $\rho^{R}(v)$ but ${\left|\rho^{R}(v)\right|}^{2}$, especially in free field environments, this time reversal step in Equation (21) is redundant. However, in waveguide environments, it can accumulate multipath components by setting the duration of the transmitted signal as the length of the block. If the duration of transmitted signal is long, the length of the block can be set as half of the duration of the transmitted signal. In the following signal processing, the length of block equals the duration of transmitted signal in the case of 0.5 ms and half of the duration of transmitted signal in the case of 1 ms. From the following analysis of simulation and experimental data in the tank, the process of iterative deconvolution has removed the blur caused by both the transmitted signal and most parts of the spread of the channel. The final time reversal processing is not very useful and can be discarded.

One can find that the ambiguity caused by the transmitted signal is removed by the iterative deconvolution and the blur caused by the multipath propagation is removed by the time reversal. Finally, we can get the reflectivity energy density of the target as a function of frequency and time-delay. This method is named the ID-TR filter. 

## 3. Simulation Results

### 3.1. The Resolution of the ID-TR Filter

#### 3.1.1. The Case in Free Field Environments

In the first simulation, the delay resolution capabilities of the proposed method and the MF are investigated. The transmitted signal is a 1-ms LFM with a central frequency of 12 kHz and a 4-kHz frequency bandwidth. The sampling frequency is 50 kHz. 

Provided that a target is placed at the position of 10 m, the received signal is simulated according to the model in Equation (1) with *α* = 1, which is an ideal target echo. In order to analysis the resolution, it is assumed that there is noise containing in the same target echo with added noise to guarantee the signal-to-noise ratio (SNR) of 20 and 0 dB, respectively. The width of the main lobe is measured by the sectional view that the amplitude is dropping to −3 dB of the maximum value. The result of MF is generated according to Equation (10). The result of generalized target reflectivity density function *ρ*(*τ*) is obtained according to Equation (18). The result of reflectivity density function of the target *c*(*v*) is obtained according to Equation (21). As shown in Figure 1, the width of main lobe of range is 0.3 m for the MF, whereas the resolution after iterative deconvolution with iteration number being 10 is approximately a delta function. The process of iterative deconvolution is implemented as iterative time reversal processing, as shown in Equation (18). With the number of iterations increasing, the ambiguity caused by the transmitted signal is more and more removed by deconvolution. The energy is concentrating and concentrating, and then finally becomes a delta function. Moreover, the first sidelobe of the MF is approximately −11 dB, whereas the first sidelobe of iterative deconvolution and the ID-TR filter is very low.

For the traditional methods, the width of the peak of the ambiguity surface reveals the resolution in delay and Doppler—the accuracy to which a target’s range and radial speed can be estimated. In active sonar, the pulsed continuous wave and linear/hyperbolic frequency modulation signal are often used in sequence [19]. When there is only one target, one can determine that target range and speed using linear/hyperbolic frequency modulation signal and pulsed continuous wave separately. Moreover, the process of deconvolution can be implemented with a long time window. This high-resolution characteristic is easy to extend to Doppler, which is demonstrated in [24]. 

#### 3.1.2. The Case in Waveguide Environments

The experimental environments and setup are shown is Figure 2, which is the laboratary waveguide. The laboratory waveguide is 1.44 m in water depth. The floor of the tank is covered with 0.22-m-thick sand, which is the sediment. The environmental parameters in the tank are as follows. The water speed is calculated as *c*_1_ = 1493 m/s by measuring water temperature and the water density is *ρ*_1_ = 1000 kg/m^3^. The speed in sediment is *c*_2_ = 1650 m/s, the density is *ρ*_2_ = 1800 kg/m^3^ and the attenuation coefficient is *α*_2_ = 0.67 dB/λ. The speed in basement is *c*_3_ = 1580 m/s, the density is *ρ*_3_ = 1800 kg/m^3^ and the attenuation coefficient is *α*_3_ = 0.8 dB/λ. The target is assumed in the range of 10 m and at depth of 0.6 m. The receiver is assumed at depth of 0.6 m. We use KRAKEN to calculate the sound pressure (Green’s function) in the frequency domain provided that the soure time series has the form $s(t)=e^{j\omega t}$ [40]:(22)$$p(r,z)\approx \frac{i}{4\rho (z_{s})\sqrt{8\pi r}}e^{-i\pi /4}\sum_{n=1}^{\infty }z_{n}(z_{s})z_{n}(z)\frac{e^{ik_{n}r}}{\sqrt{k_{n}}}$$
where *p*(*r*,*z*) is the pressure in range of *r* and at a depth of *z*, *z_n_*(*z*) is a mode shape function of *n*th propagation mode, *k_n_* is a horizontal propagation wavenumber, *z_s_* is the depth of source, and *ρ*(*z_s_*) is the density at the location of source. The Green’s functions are both the spread function of forward *h_f_*(*t*) and backward propagation channel *h_b_*(*t*) in Equation (3), which are normalized to one to guarantee different SNR. The target echo is generated using Equation (2) for a point target with reflectivity density function of the target setting as one, which spreads in time from duration of 1 to 4 ms due to the multipath propagation, as shown in Figure 3b. One finds that the target echo emerges into the noise with SNR = 0 dB in Figure 3a. We generate 200 realizations to obtain the results.

As shown in Figure 4, all the results can distinguish the two closely spaced multipath components, which are locate at the range of 10 and 10.5 m and correspond to the first two largest separated components in Figure 3b, respectively. The capability of energy concentration provide by the iterative deconvolution can achieve a resolution beyond Rayleigh limit (green line) in Figure 4a. Moreover, since the received signal is distorted by the multipath propagation, the first sidelobe located in range of 9.2 m increases to −8 dB in Figure 4a, whereas it is less than −10 dB in Figure 1a. One finds that the other multipath components can not be separated when the time delay is larger than 2 ms in Figure 3b. When the SNR is high, since the output level is −5 dB in a range between 11 and 12 m, whereas the output level of noise is only −25 dB, one find out that they can be detected using MF. However, when the SNR is lower, these components are treated as noise since they have the same output level as the noise in Figure 4b. As for ID-TR, it focuses on the strongest components with the sharpest resolution and then these components are greatly suppressed. 

### 3.2. Targets Under Noise-Limited Condition

#### 3.2.1. The Case in Free Field Environments

Under noise-limited detection conditions, the traditional way to improve the detection performance is to increase the signal energy/duration. In order to investigate its capability of anti-noise, provided that a target is at the position of 10 m, we generate 200 independent noise realizations added into the same target echo to ensure the signal-to-noise ratio (SNR) of 0 dB. For long received signal time series, one sometimes needs to divide the data into blocks using sliding (overlapping) windows to search for the target echo. Each block of data is processed using a matched filter to produce an output time series, which is also the first procedure for the ID-TR filter. The received signal is divided into blocks of equal length, which is the duration of transmitted signal in the paper. It is important that the support of *s*(*t*) is shorter than (or equal to) the length of a block and that this support is entirely included in one block [1]. This is unrealistic for the previous works due to the duration of transmitted signal is very long in order to guarantee the resolution and detection performance. The active sonar needs the transmitted signal with long duration to combat the noise [1,19,20]. However, this is a non-trivial task for the ID-TR method since it can provide a high resolution and super gain using iterative deconvolution and time reversal processing. Moreover, the iterative deconvolution can be implemented with a time window as long as possible. The target signal may be split between blocks in the procedure of the matched filter. How to recover the signal requires precise synchronization between the output signals (between different blocks). This problem is solved by the overlap-add method [41]. We cut the received signal into nonoverlapping sections of length that equal the duration of the transmitted signal *N* (*N* = *Tfs*, where *fs* is the sampling frequency). The transmitted signal and each block of the received signal are transformed into Fourier using (2N −1)-point fast Fourier transform which is expressed as *s*(*f*) and *r*_l_(*f*), where *l* is the *l*th block. The number of the point 2N −1 equals to the length of the convolution of the transmitted signal and each block of the received signal. The cross-ambiguity is implemented by the product of *r*_l_(*f*) and the conjugation of *s*(*f*) and then transform into time domain using invert fast Fourier transform. The overlap samples N − 1 of output are added in carrying out the sum to achieve the cross-ambiguity as a function of time delay. The iterative deconvolution is implemented using Equation (18) by regarding the output of matched filter as one block.

The results of 200 realizations are averaged to obtain their means, as shown in Figure 5. Figure 5a shows that an improvement of in the output SNR of MF is 11 dB, whereas the improvements of iterative deconvolution and ID-TR are approximately 60 dB in Figure 5b and 250 dB in Figure 5c, respectively. The first sidelobe of MF is approximately −11 dB. One can find that the ID-TR method has a superior gain to combat the noise. The gain is higher and higher with the number of iteration increasing and finally becomes a fixed value [24]. It has known that the array gain reflects the improvement in SNR obtained using the array, which is determined by the number of array element [42]. Using deconvolved conventional beamforming, the improvement of the array gain is about 5 dB [43]. In this paper, the iterative deconvolution is achieved in the time domain. The improvement in the array gain provided by deconvolution is limited by the array aperture, whereas the gain provided by deconvolution in the time domain is greatly improved by increasing the length of time that is 80 ms in the paper. The principle is somewhat similar to that of synthetic aperture sonar.

In the case of multiple targets, provided that there are two targets at the positions of 10 m and 11 m with a signal-to-noise ratio of 0 and 2 dB, respectively, we generate 200 independent noise realizations. As shown in Figure 6a, the relative level between the two targets is 2 dB, which is the true relative level between the two targets. However, it becomes 20 dB in Figure 6b and 100 dB in Figure 6c, respectively. Therefore, the energy is more and more focusing on the strongest target with the number of iterations increasing. In fact, the process of deconvolution is achieved in Equation (18) by the iterative time reversal method that the cross-ambiguity is reversed in time and then retransmitted numerically to remove the ambiguity caused by the transmitted signal. A time reversal method is implemented in Equation (21) to remove the blur caused by the channel. Therefore, the ID-TR method is an iterative time reversal processing. The characteristic of iterative time reversal processing is that it can focus on the strongest target finally [28,31,32]. However, the traditional iterative time reversal is implemented using time reversal array to remove the blur caused by the channel. 

#### 3.2.2. The Case in Waveguide Environments

The received signal is the same as that in Figure 3a with SNR = 0 dB. We generate 200 independent noise realizations and average to obtain the results in Figure 7. Due to the multipath propagation, the received signal is distorted and then the detection performance is degrated as compared to Figure 5. Figure 7a shows that the improvement in the output SNR of MF becomes 6 dB with a performance degradation of 5 dB compared to the ideal case in Figure 5a. The improvements in the output SNR of iterative deconvolution and ID-TR method is 30 dB and 100 dB. Since the last equation in Equation (11) is not ture in waveguide environments, the performance degradation of the ID-TR method is more serious than that of the MF. However, it still has a superior gain to combat the noise. Although the multipath components after 2 ms in Figure 3b are greatly suppressed compared to the strongest component in range of 10 m, they obviously present as peaks in a range between 11 and 12 m in Figure 7c due to the superior anti-noise capability of the ID-TR method.

In the multiple targets case, we assumed that there are two point targets in the range of 8 and 10 m and at depth of 0.6 m, the SNR of which is 0 and 2 dB, respectively. Due to the coupling between the two targets [44,45], the target echo in range of 10 m in Figure 8b is different with that in Figure 3b. The target echo includes the forward propagation from the source to target, the target reflectivity function, and the backward propagation from the target to the receiver. Therefore, the echoes of the two targets are different from each other for the same target but at different positions of waveguide due to the forward propagation from the source to target, and the backward propagation from the target to the receiver are different. We generate 200 independent noise realizations and average to obtain the results in Figure 9.

Due to the fact that the two targets are close to each other and the SNR is low, it is not easy to find out the multipath components for both targets. Compared to the two targets in free field environments, the performance degradation of MF is 7 dB since the peak to side ratio is from −13 dB in Figure 6a to −6 dB in Figure 9a. In waveguide environments, the noise background becomes more fluctuant than MF after using iterative deconvolution, as shown in Figure 7 and Figure 9. However, the improvements in the output SNR of iterative deconvolution and ID-TR method is still 25 dB and 100 dB. Although the performance degradation of the ID-TR method is more serious than that of the MF, it still has a superior gain to combat the noise. The performance degradation is similar as the case for a single target in Figure 7.

## 4. Experimental Results

### 4.1. The Case of an Extended Target

As analyzed above, the implementation of the ID-TR filter is a process of iterative time reversal. It has been demonstrated that the time reversal has a superior ability to suppress the reverberation [28,29,30,31,32,33,34]. In order to investigate the performance of the ID-TR filter under reverberation condition, two experiments have been carried out in a tank. Target detection is more difficult in a reverberation/clutter-limited environment, where the clutters are often target like, and detection might not improve by increasing the number of iterations (in contrast to the noise limited case). It is noted that the reverberation in real oceans is highly non-Gaussian, and clutter varies significantly depending on the bottom and is very difficult to model either analytically or numerically. For reverberation/clutter-limited environments, one is often forced to use real data for performance evaluation [20]. This paper studies the performance of the ID-TR method in a reverberation-limited environment using experimental data in a tank.

The experimental environments and setup are as shown in Figure 2. The laboratory waveguide is 1.44 m in depth, 1.2 m in width and 14 m in length. The source-receiver array (SRA) of 32 elements is vertically deployed from the surface to the floor. The first transducer is 0.04 m apart from the surface. The space between elements is 0.04 m with array aperture equaling to the water depth. Underwater sound absorbing materials stick on the three sides of the tank. The other side is a steel plate at a range of 13 m from SRA, which is used for producing waves. The floor of the tank is covered with 0.22-m-thick sand, which is the sediment. In the first experiment, some bricks are placed on the sand to generate strong reverberation, which are settled in the softer sand. In order to exclude the high scatter near the source, the data of analysis is chosen in the range between 2 and 14 m. The target is an air-filled steel cylindrical shell with a diameter of 0.21 m and a length of 0.51 m in the first experiment, which is approximately placed in the range of 8.5 m on the seafloor. An exact definition of extend targets has not existed at present, but it usually regards the target whose typical size is comparable to the acoustic wavelength as an extended one [33]. The transmitted signal is a 1-ms LFM with a central frequency of 12 kHz and a 4-kHz frequency bandwidth. The SRA transmits a broadside ping, that is, each source element transmits LFM with equal amplitude.

We chose the signal received by the 15th array element placed at the depth of 0.6 m to achieve the target detection and range estimation. The number of iterations is 8. The results are shown in Figure 10. The noise level is suppressed by more than 20 dB by using iterative deconvolution, whereas the reverberation level in the range of 4 m is only suppressed by less than 3 dB in Figure 10b. We ignore the strong returns in the range of 13 m, which are caused by the steel plate. By removing the blur caused by the spread of the channel, one finds that the reverberation is suppressed by more than 2 dB in the range of 4 m and the noise is suppressed by more than 50 dB by the time reversal processing in Figure 10c. The processing gain of ID-TR is more than 70 dB. One finds that it is difficult to suppress the reverberation with the similar level as the target.

### 4.2. The Case of Ideally Resolved and Extended Targets

For the second experiment, the targets are two extended targets that are the same as the steel cylindrical shell used in the first experiment. One is place at the bottom of tank is in range of 5.9 m and the other is in range of 7.9 m and at a depth of 0.83 m. The transmitted signal is a 0.5 ms LFM with a central frequency of 18 kHz and a 4 kHz frequency bandwidth. The 12th element of SRA at depth 0.48 m acts as a source to transmit the signal. The bricks are removed.

The receiver waveform after MF processing is shown in Figure 11a for a target range of 3 to 12 m. One observes many scattered returns if a single element is used. The high-level returns are suppressed by 5 dB by using iterative deconvolution with the iteration number being 3. The number of iterations is small to guarantee the weak target not to be simultaneously suppressed in Figure 11b. The two targets can be simultaneously detected using the ID-TR method in Figure 11c. However, due to the duration of transmitted signal is short and the size of target is big, two peaks simultaneously present in the range of 6 m and the mainlobe of the two targets are widened.

We average all the results of the array elements to achieve the target detection and range estimation. Since the weak target can be greatly suppressed using ID-TR, the number of iterations is 3. The results are shown in Figure 12. The receiver waveform after MF processing is shown in Figure 12a, the two targets are at a range of 6 and 8 m, respectively, which is a similar result by using the ID-TR method with a single element in Figure 11c. Due to the short duration of the transmitted signal and a big size of the target, one finds that two peaks are simultaneously present at the range of 8 m using a matched filter in Figure 12a. By removing the ambiguity caused by the transmitted signal and the channel, one observes that two peaks combine into one peak in Figure 12c. The weak target is suppressed more than 2 dB in Figure 11c. Therefore, the number of iterations should be set to a smaller number in practical application. The depth of the tank is only 1.44 m, and then the multipaths are very strong. Moreover, the size of the target is very big compared with the tank. The time reversal processing in Equation (21) is actually accumulated all the components caused by the multipaths and the extended target, which can also be demonstrated both in Figure 10 and Figure 11.

## 5. Discussion

MF is widely used in active sonar because of its simplicity and ease of implementation. The major drawback of the matched filtering is that its processing gain and resolution are limited by the transmitted signal. Modern high-gain active sonar uses wideband and/or long duration signals to improve the signal-to-reverberation ratio. However, it may not be suited to the use of signals with a long duration since underwater environments exhibit rapid time variation. We put forward an ID-TR method that can achieve super-resolution and a high processing gain in the condition of smaller bandwidth and shorter time duration.

The underlying physical process of wave propagation would be unchanged if time is reversed. This is a special case of the Richardson-Lucy algorithm that the blur function is one-dimensional and time-invariant. Therefore, the iteration deconvolution can be implemented by the iterative time reversal processing. Different with the conventional time reversal method, the time reversal processing is implemented not in space but in the time domain. Compared to the general Richardson-Lucy algorithm, the special case has a superior ability of anti-noise and a rapid speed of convergence, which will be discussed in detail in the future.

For an “ideal” case that the received signal is a time-delay of the transmitted signal, the sidelobe of the MF in this paper is −11 dB and then the gain of MF is 11 dB. The gain of ID-TR is 250 dB when the number of iterations is 10 with a resolution as a delta function. In a waveguide environment with strong multipaths, the received signal is greatly distorted due to the superposition of multiple components of target echoes from different paths. The gain of MF is degraded from 11 to 6 dB, whereas the gain of ID-TR decreases from 250 to 100 dB. Although the performance degradation of the ID-TR is more serious than that of the MF, the ID-TR still has enough gain to combat the noise. In fact, the procedure of iterative deconvolution is implemented by iterative time reversal processing. Therefore, with the number of iterations increasing, the energy is more and more concentrating on the strongest target when more than two targets present. However, the weak target can still emerge out from the noise due to the superior ability of anti-noise provided by the ID-TR method. In the practical application, the number of iterations should be set to a smaller number in order to detect the weak target. It is not suitable to apply in a high clutter environment and we are trying to solve it. It is possible that it can be solved in the future. From the analysis of simulation and experimental data in the tank, the process of iterative deconvolution has removed the blur caused by both the transmitted signal and most parts of the spread of the channel. The final time reversal processing is not very useful and can be discarded.

When the reverberation is dominant, the MF output often contains many high-level returns of comparable levels as the target echo, which is known as clutter and can confuse the target. For the traditional method, clutters are the reasons for a high probability of false alarm in the case of a small number of snapshots (small number of pings). Using a single snapshot and a single receiver, the ID-TR can differentiate the target echo from the clutter to a certain degree due to the capability of reverberation suppression provided by the iterative time reversal, which is demonstrated by the experimental data in a tank. Moreover, the ID-TR is very easy to be implemented by FFT and IFFT. However, the experimental tank is not a strong reverberation environment, whereas it is more like a noise-limited detection condition. Therefore, the ability of the anti-reverberation of ID-TR has not been demonstrated. This is one of the works that should be done in the future. 

## 6. Conclusions

In this paper, we put forward an ID-TR method, which can simultaneously remove the blur caused by the transmitted signal and the spread function of channel. Therefore, the range resolution is approximately a delta function even though the transmitted signal is a LFM with a central frequency of 12 kHz and a 4-kHz frequency bandwidth. Moreover, the peak-signal power is becoming larger and larger with the number of iteration increasing. Therefore, the peak-to-sidelobe level is improved. The simulated result shows that the noise reduction can achieve 250 dB in the “ideal” free field environment and 100 dB in a strong multipaths waveguide environment if a 1-ms linear frequency modulation with a 4-kHz frequency bandwidth is transmitted and the number of iteration is 10. The results of the experiment in a tank show that the noise level is suppressed by more than 70 dB and the reverberation level is suppressed by 3 dB in the case of a single target and the iteration number being 8.

## 7. Patents

The patent named an iterative deconvolution-time reversal method for target detection and range estimation is accepted with number 202010123948.1 in China.

## Figures and Tables

**Figure 1 sensors-20-02844-f001:**
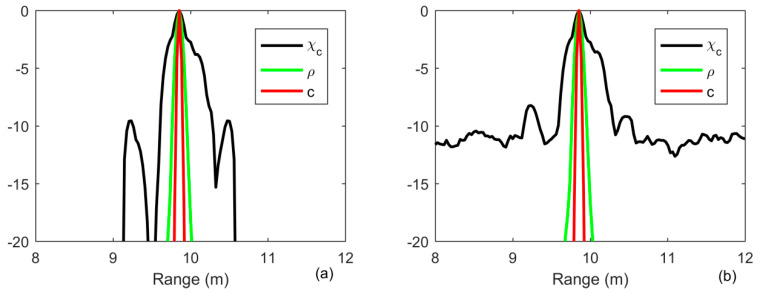
Output of the matched filter (black line), iterative deconvolution with iteration being 10 (green line) and the iterative deconvolution-time reversal filter (red line) in free field environments (**a**) SNR = 20 dB; (**b**) SNR = 0 dB.

**Figure 2 sensors-20-02844-f002:**
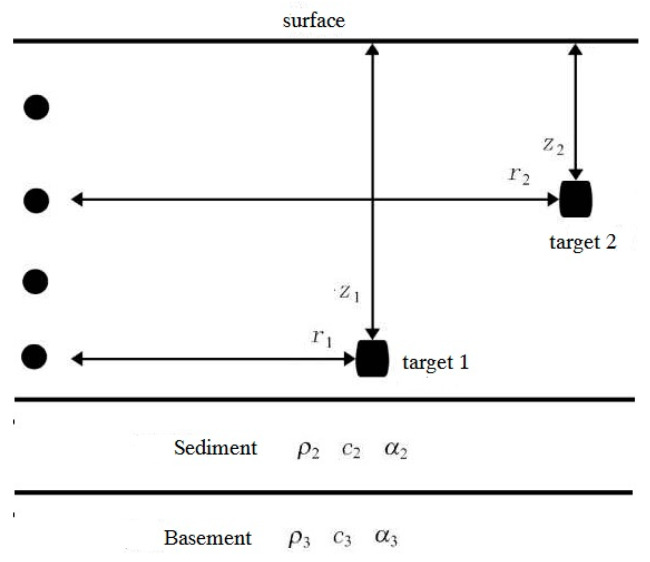
Experimental setup and ocean’s acoustical environment.

**Figure 3 sensors-20-02844-f003:**
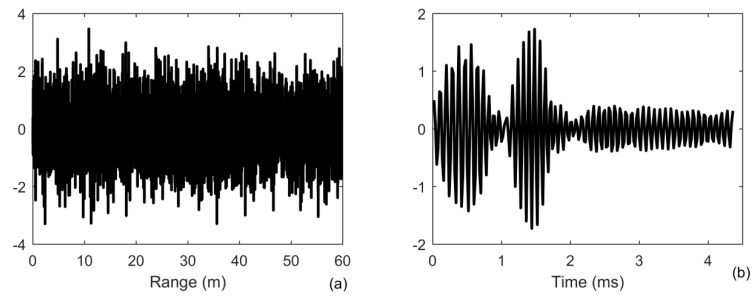
Signal received by the receiver (**a**) as a function of range; (**b**) target echo in range of 10 m.

**Figure 4 sensors-20-02844-f004:**
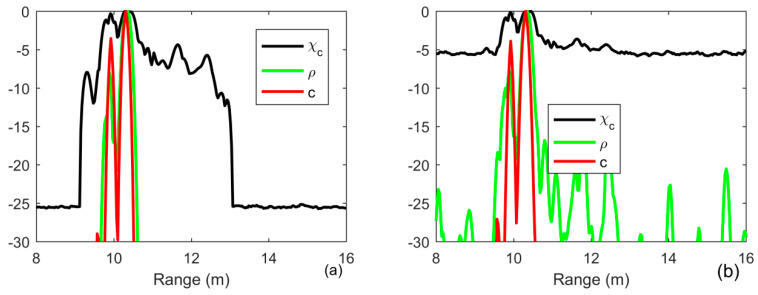
Output of the matched filter (black line), iterative deconvolution with iteration being 10 (green line) and the ID-TR filter (red line) in waveguide environments (**a**) SNR = 20 dB; (**b**) SNR = 0 dB.

**Figure 5 sensors-20-02844-f005:**
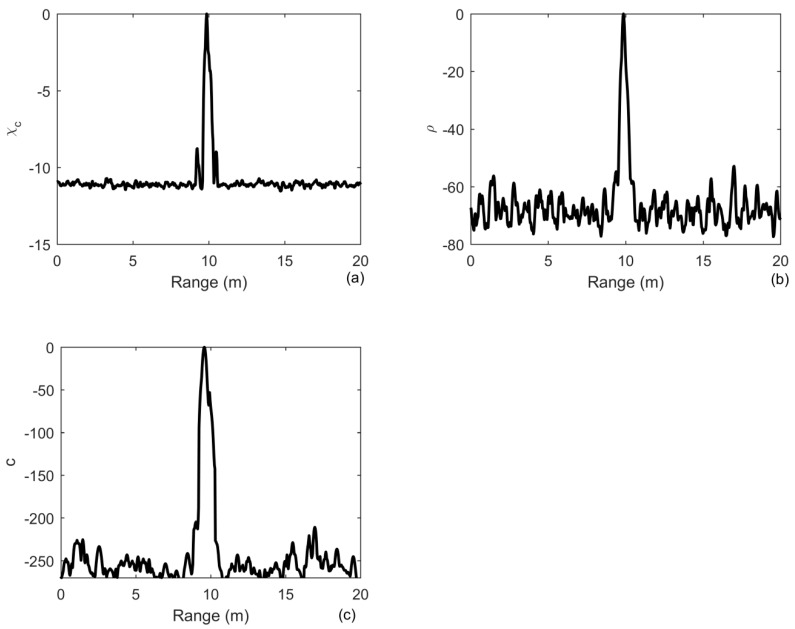
Output of: (**a**) the matched filter; (**b**) iterative deconvolution with iteration being 10; (**c**) the ID-TR method for a signle target under the noise-limited detection condition in free environments.

**Figure 6 sensors-20-02844-f006:**
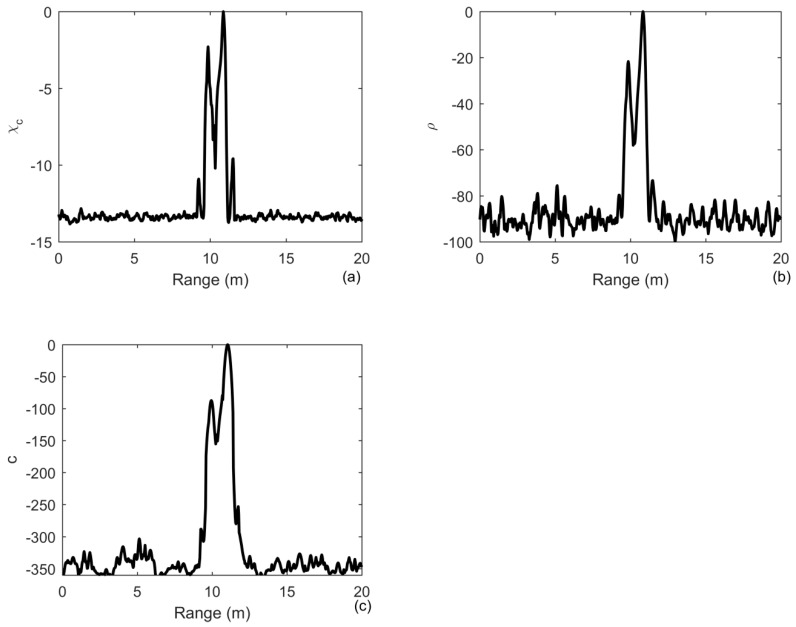
Output of: (**a**) the matched filter; (**b**) iterative deconvolution with iteration being 10; (**c**) the ID-TR method for two targets under the noise-limited detection condition in free environments.

**Figure 7 sensors-20-02844-f007:**
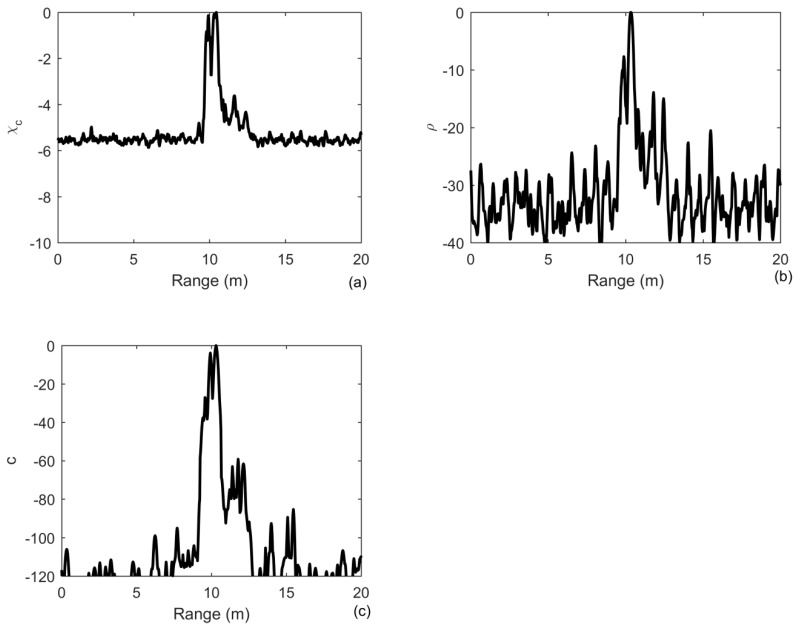
Output of: (**a**) matched filter; (**b**) iterative deconvolution with iteration being 10; (**c**) the ID-TR method under the noise-limited detection condition in waveguide environments.

**Figure 8 sensors-20-02844-f008:**
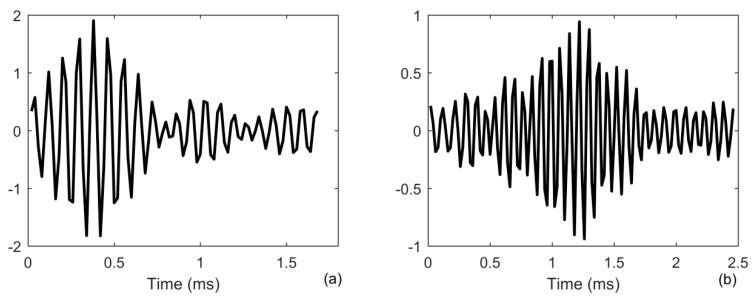
Target echo for multiple targets in waveguide environments: (**a**) in range of 8 m; (**b**) in range of 10 m.

**Figure 9 sensors-20-02844-f009:**
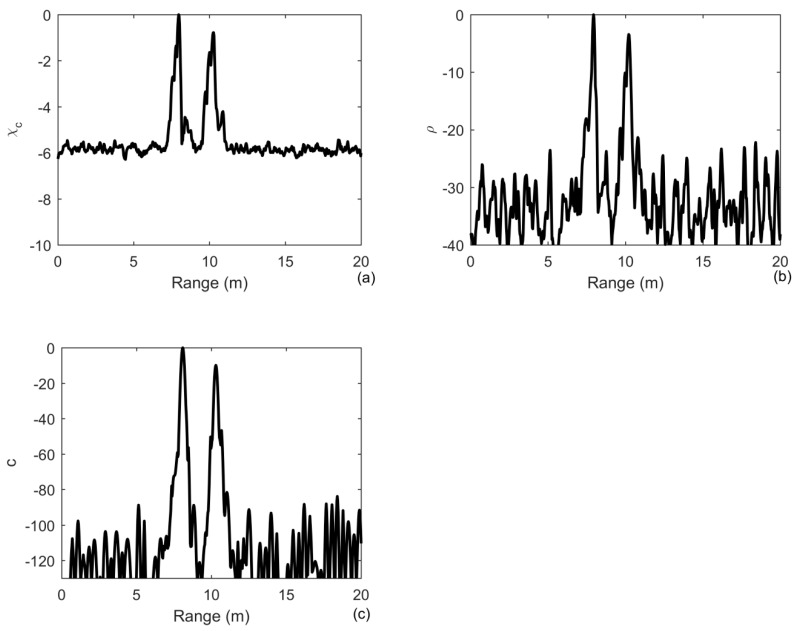
Output of: (**a**) the matched filter; (**b**) iterative deconvolution with iteration being 10; (**c**) the ID-TR method under the noise-limited detection condition for multiple targets in waveguide environments.

**Figure 10 sensors-20-02844-f010:**
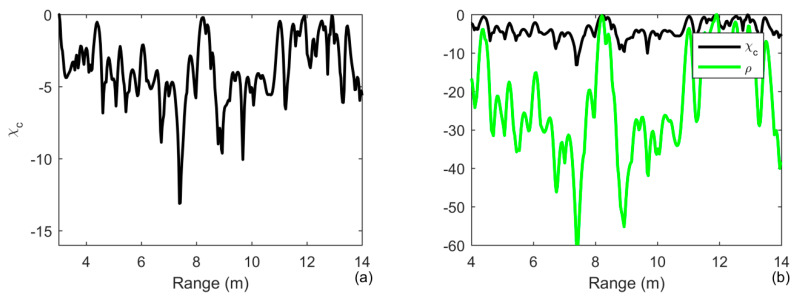
Output of (**a**) matched filtering (MF); (**b**) the comparison of MF and Iterative deconvolution with iteration being 8; (**c**) the comparison of Iterative deconvolution with iteration being 8 and ID-TR filtering under the reveberation-limited detection condition by choosing the array element placed at a depth of 0.6 m in the case of an extended target.

**Figure 11 sensors-20-02844-f011:**
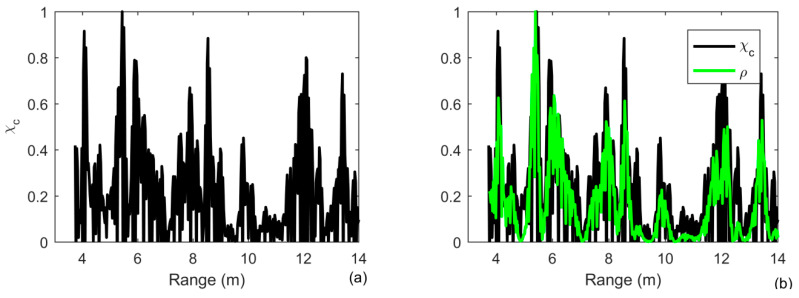
Output of (**a**) MF, (**b**) the comparison of MF and iterative deconvolution with iteration being 3; (**c**) the comparison of iterative deconvolution with iteration being 3 and ID-TR filtering under the reveberation-limited detection condition in case of ideally resolved and extended targets by choosing the array element placed at depth of 0.6 m in the case of an extended target.

**Figure 12 sensors-20-02844-f012:**
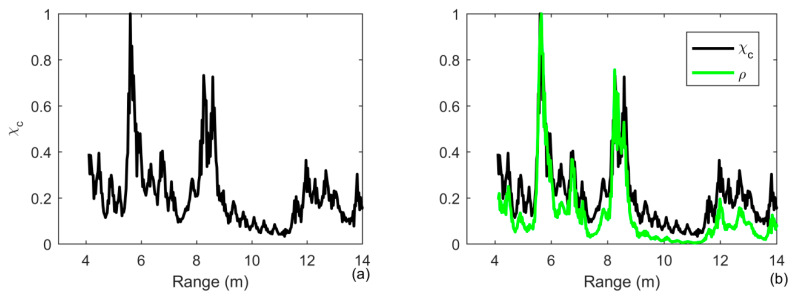
Output of (**a**) MF; (**b**) the comparison of MF and Iterative deconvolution with iteration being 3; (**c**) the comparison of Iterative deconvolution with iteration being 3 and ID-TR filtering under the reveberation-limited detection condition in case of ideally resolved and extended targets by averaging the results of all array elements in the case of an extended target.

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
