# Peer review of "An Iterative Deconvolution-Time Reversal Method with Noise Reduction, a High Resolution and Sidelobe Suppression for Active Sonar in Shallow Water Environments"

_sensors, 2020, doi:10.3390/s20102844_

Round 1

Reviewer 1 Report

  1. This paper was written well and professionally. A novel algorithm for active sonar detection was proposed to achieve better resolution and anti-noise capability. Just a few minor changes are suggested in the following.
  2. Line 56: “Therefore, the difficulty of detecting a target is great greatly exemplified when a low Doppler target is situated in shallow water environments,” “great” should be removed.
  3. Line 62: “The object of this paper is” should be changed to “The objective of this paper is”.
  4. The equations of (6), (20), and (21) are distorted. Please check.

Author Response

Dear professor, I am so happy to receive your compliment and encouragement. I will carefully revise my article according to your suggestions. Thank you for giving me so much valuable advice. 1、Question: Line 56: “Therefore, the difficulty of detecting a target is great greatly exemplified when a low Doppler target is situated in shallow water environments,” “great” should be removed. Answer: Thanks for the reminding, we have removed “great”. . 2、Question: “The object of this paper is” should be changed to “The objective of this paper is”.. Answer: This sentence has been modified. 3、Question: The equations of (6), (20), and (21) are distorted. Please check. Answer: Regarding Comments 3), these equations have been modified.

Reviewer 2 Report

In the article iterative deconvolution-time reversal method for target detection and range estimation is presented. It is interesting approach for sonar data filtering in the noisy environment. Article is well organized and written. Final conclusion are well substantiated. Bibliography is proper but a little bit old. Some minor remarks:

  1. Eq.6 looks be incomplete
  2. Eq.21. is unreadable
  3. Line 261 “labortary”

Author Response

Dear professor, I am so happy to know that you are interested in my study. Thank you for your compliment and encouragement. I will carefully revise my article according to your suggestions. Thank you for giving me so much valuable advice.

1、Question: Line 261 “labortary”

Answer: Thanks for the reminding, we have modified as “laboratory”.

2、Question: Bibliography is proper but a little bit old.

Answer: Regarding Comments 5), we have added many recent published papers, which are shown in red color in the section of references.

  1. Liu, X.; Sun, C.; Yang, Y.; Zhuo, J. Low sidelobe range profile synthesis for sonar imaging using stepped-frequency pulses. IEEE Geosci. Remote. Sens. Lett. 2017, 14, 218-221.
  2. Zou, J.; Yang, R.; Luo, S.; Zhou, Y. Range sidelobe suppression for OFDMintegrated radar and communication signal. Eng. 2019, 2019 ,7624-7627.
  3. Dong, Y. Implementable phase-coded radar waveforms featuring extra-low range sidelobes and Doppler resilience. IET Radar Sonar Navig. 2019, 13, 1530-1539.
  4. Hua, G.; Abeysekera, S. S. Receiver Design for Range and Doppler Sidelobe Suppression Using MIMO and Phased-Array Radar. IEEE Trans. Signal  Proces. 2013, 61, 1315-1326.
  5. Zhou, S. H.; Liu, H. W.; Wang, X., Cao, Y. MIMO Radar Range-Angular-Doppler Sidelobe Suppression Using Random Space-Time Coding. IEEE T. Aero. Elec. Sys. 2014, 50, 2047-2060.
  6. Berestesky, P.; Attia, E. H. Sidelobe leakage reduction in random phase diversity radar using coherent CLEAN. IEEE T. Aero. Elec. Sys. 2019, 55, 2426-2435.
  7. An, Q.; Hoorfar, A.; Zhang, W. J.; Li, S. Y.; Wang, J. Q. Range coherence factor for down range sidelobes suppression in radar imaging through multilayered dielectric media. IEEE Access. 2019, 7, 66910-66918.
  8. Li,, S.; Amin, M., An, Q., Zhao, G., Sun, H. 2-D Coherence Factor for Sidelobe and Ghost Suppressions in Radar Imaging. IEEE Trans. Antennas Propag. 2020, 68, 1204-1209.
  9. Yazici, B.; Xie, G. Wideband extended range-doppler imaging and waveform design in the presence of clutter and noise. IEEE Trans. Inform Theory. 2006. 52, 4563-4580.
  10. Yu, L.; Ma, F.; Lim, E., Cheng, E.; White, L. B. Rational-orthogonal-wavelet-based active sonar pulse and detector design. IEEE J. Ocean. 2019, 44, 167-178.
  11. Dalitz, C.; Pohle-Fröhlich R.; Michalk, T. Point spread functions and deconvolution of ultrasonic images. IEEE Trans. Ultrason. Ferr. 2015, 62, 531-544.
  12. Wu, J.; Aardt J. A. N.; Asner, G. P. A comparison of signal deconvolution algorithms based on small-footprint liDAR waveform simulation. IEEE Trans. Geosci. Remote. 2011, 49, 2402-2414.
  13. Kirk, B. H.; Martone, A. F.; Sherbondy, K. D.; Narayanan, R. M. Mitigation of target distortion in pulse-agile sensors via Richardson-Lucy deconvolution. ELECTRONICS LETTERS 2019, 55, 1249-1252.

3、Question: Eq.6 looks be incomplete

Eq.21. is unreadable

Answer: Regarding Comments 6), these equations have been modified.

Reviewer 3 Report

To be clear from the beginning of this Review, I am no expert in this field but the title and abstract looked very promising and I was very keen to learn more about what promises to be an important development. After carefully reading the paper, I would say it needs substantial revisions.

These revisions are necessary to properly explain the method (especially to non-specialists) and the details of the experiment (in particular the role of target surface and volume scattering in the echoes observed). Some of the equations are incomplete, at least in the PDF version I have reviewed, and there are many questions that need addressing. And there is no Conclusion section at all.

Here are my comments, by order of appearance in the paper.

The title is attractive, with the promise of high resolution and anti-noise. I would however suggest replacing “super capability” with quantitative arguments (e.g. noise reduction and increase in resolution offered by this approach).

The Abstract focuses on the methods, which is justified, but it should also quantify the improvements they bring, or at least mention the conditions of applications (e.g. bandwidths, ranges, resolution improvements).

Check English grammar throughout. Most of us are not native speakers, so I fully understand the issues we face when writing in a foreign language. Some corrections are easy to make through proof-reading (e.g. “method … method …” in the Abstract, “great greatly” lines 56/57). Others can get in the way of clear and unambiguous communication, e.g. “but in a sequent time” (line 47),

Great bibliography, varied and wide-ranging.

Line 46: how do “scaled versions of a single waveform” help? Are they scaled in amplitude, in time or in frequency?

Lines 74-75: define what “a super processing gain” actually is, ideally with numbers. Line 76: constrain “to a certain degree” with numbers too.

Line 96: “dependent” not “depended”. Same on line 138.

Line 98: “should take into account” rather than “should be taken into account”.

Line 99: can this approach to sound propagation account for propagation into/out of the seabed as well? (I can understand if this is not possible for a first approach, as it will quickly become complicated to look at all possible cases).

Line 119: equation (6) is incomplete.

Line 154: please add a reference to the original paper by Richardson and Lucy.

Lines 156-160: please detail why the Csiszar discrimination is the most appropriate.

Line 172: explain what is the Kuhn-Tucker condition (including a reference to a relevant paper) and why it must be satisfied.

Line 179: “It [was] proved that the Kullback discrimination …”. Add reference to who proved it. Why do we need about that? How does it evolve from the Csiszar discrimination? To be understood by a wider readership and people not that familiar with these techniques (like this reviewer), it is important to clearly justify all these steps.

Line 189-190: in the same vein, explain why “the special case has a superior ability of anti-noise and a rapid speed of convergence”. How superior is superior? Is it possible to quantify it at this stage? (I am not sure it can be quantified, but this “superior ability” must be justified nonetheless).

Line 205: equation (20) looks incomplete.

Line 211: check the formatting of equation 21 as I am not sure it shows properly at all.

Line 222: explain why the “final time reversal processing” is considered to be “not very useful”.

Line 237: “SNR of 20 dB and 0 dB respectively”. I do not understand “respectively”: is that two SNR cases, for the target echo, or SNRs for two different echoes (one from the target and the other one from where?).

Figure 1: the Y-axis is presumably in relative dB, but the legends show the iterative deconvolution is represented in green (with label “rho”) and ID-TR in red (with label “c”). This is confusing, as the nomenclature identifies them as “reflectivity density function of the target” and its generalised form, respectively.

Line 262: is the sediment cover of 0.22 m included in the 1.44-m depth or is the 1.4 m only for the water column, as would seem logical from the text?

Line 266: what is meant by “basement”? Is that the bottom of the tank, or an additional layer? What is it made of?

Figure 2: is the image at scale? (to know the ranges to targets). Line 267 says “the target is assumed in the range of 10 m and at a depth of 0.6 m”, but this does not correspond to this figure (see line 337 for what the ranges could be to each target: the image is then definitely not to scale). Or could it be for the ranges in line 370 (8 m and 10 m)? This is not clear at all.

Line 294: “focuses” and not “fouces”

Line 299: “noise” and not “nosie”

Line 317: explain what the “overlap-add method” is (reference [27] is fine, but a quick summary of the approach is needed for the reader of your article).

Lines 322-323: these improvements are the numbers that should be presented in the Abstract, as they are very important and should be presented early on in the article.

Line 363: is “behide” a misspelling of “behind” or “beside”?

Figure 8: the reader will need converting the times (in ms) to distances travelled in water (i.e. ranges). Is that possible to add that as separate labels? Please indicate the ranges of 8 m and 10 m exactly, e.g. with arrows pointing to where they are. An important component of target echo can also be surface waves and volume scattering within the targets. As there is NO information about the targets, it is impossible to assess what Figure 8 tells us exactly about the target echoes. Why are these two target echoes dissimilar, by the way? Are the targets identical or not (I do not remember it being explained in the description of the experiment)? Arguably, this is still a simulation, but target characteristics should be made clear early on.

The following section 4 (on the experiment) has more information on the targets (“steel cylindrical shell”, but is it empty or filled, and with what? If so, what is its wall thickness?). What is the purpose of the steel plate at 13- m range? Where are the bricks placed and are they proud or do they settle in the softer sand because of their weight (and to what depth)? I would point the authors to work by A. Tesei and collaborators, often published in J. Acoust. Soc. Am., to see examples of why this is important, and what it can add to the actual echoes from each type of target.

Line 422: why choose the 15th array element? How does it differ from the 14th or 16th element?

Line 432: which targets? The cylindrical shells?

The Discussion mentions that “multipaths are strong”, and it would be nice to see an explanation of how time-gating can mitigate the role of the steel plate at far range in the overall acoustic field.

Section 5 (Discussion) is followed with another Section 5 (Patents) associating this work with a single patent, but there is no Conclusion. What is the role of informing the readers of this patent? It can potentially be useful but it is not explained. And the absence of Conclusion is rather puzzling.

In short, there are enough revisions to make before this paper can be considered again.

Round 2

Reviewer 3 Report

Thanks for engaging thoroughly with the review: I am looking forward to seeing this paper published.

This manuscript is a resubmission of an earlier submission. The following is a list of the peer review reports and author responses from that submission.